# Judo Injuries Frequency in Europe’s Top-Level Competitions in the Period 2005–2020

**DOI:** 10.3390/jcm10040852

**Published:** 2021-02-19

**Authors:** Wiesław Błach, Peter Smolders, Łukasz Rydzik, Georgios Bikos, Nicola Maffulli, Nikos Malliaropoulos, Władysław Jagiełło, Krzysztof Maćkała, Tadeusz Ambroży

**Affiliations:** 1Faculty of Physical Education & Sport, University School of Physical Education, 51-612 Wroclaw, Poland; wieslaw.judo@wp.pl (W.B.); krzysztof.mackala@awf.wroc.pl (K.M.); 2European Judo Union, 1200 Vienna, Austria; smolderspeter@skynet.be; 3Institute of Sports Sciences, University of Physical Education, 31-571 Krakow, Poland; tadek@ambrozy.pl; 4Euromedica-Arogi Rehabilitation Clinic, 54301 Thessaloniki, Greece; bikosg77@yahoo.gr; 5Department of Orthopaedics, School of Medicine, Surgery and Dentistry, 89100 Salerno, Italy; n.maffulli@qmul.ac.uk; 6Centre for Sports and Exercise Medicine, Queen Mary, University of London, London E1 4DG, UK; contact@sportsmed.gr; 7Institute of Science and Technology in Medicine, Keele University School of Medicine, Stoke on Trent E1 4DG, UK; 8Sports and Exercise Medicine Clinic, 54639 Thessaloniki, Greece; 9Sports Clinic, Rheumatology Department, Barts Health NHS Trust, London E1 4DG, UK; 10Department of Sport, Gdansk University of Physical Education and Sports, 80-336 Gdansk, Poland; wjagiello1@wp.pl

**Keywords:** sports injuries, judo, frequency, prevalence, type

## Abstract

Background: The present study assesses the frequency of injury in Europe’s top-level judokas, during top-level competitions, and defines risk factors. Methods: The members of the EJU Medical Commission collected injury data over the period of 2005 to 2020 using the EJU Injury Registration Form at Europe’s top judoka tournaments. Results: Over the 15 years of the study, 128 top-level competitions with 28,297 competitors were included; 699 injuries were registered. Of all competitors, 2.5% needed medical treatment. The knee (17.4%), shoulder (15.7%), and elbow (14.2%) were the most common anatomical locations of injury. Sprains (42.2%) were by far the most frequent injury type, followed by contusions (23.1%). Of all contestants, 0.48% suffered an injury which needed transportation to hospital. There was a statistically significant higher frequency of elbow injuries in female athletes (*p* < 0.01). Heavy-weight judokas suffered a remarkably low number of elbow injuries, with more knee and shoulder injuries. Light-weight judokas were more prone to elbow injuries. Conclusions: We found there was a low injury rate in top-level competitors, with a greater frequency of elbow injuries in female judokas. During the 15 years of injury collection data, an injury incidence of 2.5% was found, with a remarkable high injury rate in the women’s −52 kg category, and statistically significantly more elbow injuries in women overall.

## 1. Introduction

Judo is a highly technical sport based on the principle of “maximum efficiency with minimum effort” [1]. A judo fight starts with the opponents both standing, attempting to throw each other off balance. After a throw, judokas transition to ground-fighting, the so-called “ne-waza” [2].

The fighting environment consists of constant changes of actions with applications of different movement structures [3]. The more athletes train and compete, the greater the range of powerful throwing techniques they are exposed to, and the chance of injury [4,5,6,7,8]. The frequency and number of injuries, as well as the severity of the injury, influences further training and competitions [9].

Recent studies analysing the frequency and type of injury in judo are available [10,11,12,13,14,15]. The rate of injury ranges between 12.3% and 30% [16,17]. Data on a large population (all ages and levels of performance) of French judokas during contests showed an injury incidence slightly above 1.1% [18].

The aim of the present investigation was to assess the frequency of injury in Europe’s top-level judokas during high-level contests.

## 2. Materials and Methods

### 2.1. Subjects

Data were collected from a group of 26,862 high-performance judokas (15,571 men and 11,291 women) aged between 19 and 35 years in all judo weight categories competing in 128 international tournaments under the auspices of the European Judo Union (EJU), including European Judo Championships, in the period between 2005 and 2020. The participants were informed of the protocol and procedure of the EJU Injury registration form. Athletes signed the informed consent form. The EJU Injury registration form was approved by the Medical Commission of EJU. The study was approved by the Bioethics Committee at the Regional Medical Chamber (No. 287/KBL/OIL/2020).

### 2.2. Study Design

All relevant information was obtained using the questionnaire controlled and supervised by the European Judo Union (EJU) Medical Commissioner present at each competition. When judokas were injured, they were asked to complete this questionnaire and provide relevant information, with the help of the local medical staff and the EJU medical commissioner present. “Minor” injuries, such small nose bleeds or skin abrasions, which do not influence the athlete’s performance in any way, were not counted. “Serious” injuries were defined as injuries so severe they needed transportation of the athlete to hospital. The Cronbach’s alpha (=0.71, which is considered an acceptable value) was used to assess the validation of the EJU Injury registration form.

### 2.3. Data Acquisition

In the present study, an injury was defined as the physical condition which necessitated an intervention or medical advice by the medical team present at the judo tournament or a visit to the hospital. After each medical intervention, the injured athlete or the medical staff was asked to complete the medical form. The first part of the form asked the judokas to give general information, including their gender and weight category. In the second part, the medical staff was able to collect data on the anatomical location of the injury, type of injury, structure involved, side of the lesion, and whether the judoka was allowed to continue the fight. The diagnosis of the injury was always filled in on the medical form by the treating physician, either the team doctor or the physician of the local medical team. To ensure the privacy of the injured athlete, individual names were never mentioned. When the judoka had to be transferred to hospital, the injury was defined as “serious”.

### 2.4. Statistical Analysis

The following variables were examined: gender, weight class, body regions, type of injury, and whether or not the athlete had to be transported to hospital, and the injury frequency of each body region was calculated. The Student’s *t*-test and chi-square test were used to evaluate the differences in incidence rates of specific injuries regarding the sex and weight categories. Statistical significance was set at *p* = 0.05. Data were analyzed using Microsoft Windows SPSSWINN 21.0.

## 3. Results

Of the 699 injured judokas, 384 (54.9%) were men and 315 (45.1%) were women. Overall, 2.5% of all participating judokas needed medical assistance, with no significant difference between men and women (*p* > 0.05).

Table 1 presents the anatomical location of all injuries broken down into smaller units. A total of 696 Injury Registration Forms were filled in correctly in terms of anatomical location. The most frequently injured location was the knee (17.4%), closely followed by shoulder (15.7%) and elbow (14.2%). If we compare the three most frequently occurring anatomical locations in both genders, we find that there is no statistically significant difference in shoulder and knee injuries. However, women had statistically significant more elbow injuries when compared to men (*p* < 0.01).

Table 2 shows the type of injury which occurred during tournaments. On this subject, we received 695 correctly filled-in forms. The highest percentage rates were sprain (42.2%), occurring with equal frequency in men and women judokas. Soft tissue contusions were second, with an incidence of 23.1%, again occurring with equal frequency in men and women. Men experienced significantly more bleeding episodes than women. We caution that minor nose bleeds and superficial skin lesions were not counted, since they were not considered an injury and did not necessitate medical intervention. Sixty-one luxations occurred, 36 located at the shoulder and 10 at the elbow joint. Unconsciousness after strangling and choking techniques constituted a small percentage of the total number of injuries (6.8%).

Injuries were also classified according to their severity and the inability to continue fighting. A serious injury was defined as an injury which required transport to hospital. In the time span of our investigation, a total of 136 judokas suffered a serious injury, and 0.48% of all competitors needed transport to hospital. Of these 136 judokas, 72 were male and 64 were female (*p* > 0.10). The most common location of serious injuries was the shoulder: 36 judokas had to be transferred to hospital because of a shoulder injury. Hence, one-third of all judokas experienced serious shoulder injuries, and almost 26.5% of all serious injuries involved the shoulder. Thirty-two judokas experiencing elbow injuries were transferred to hospital. Of the elbow injuries, 32.3% were classified as serious, and 23.5% of all serious injuries were located at the elbow joint. There was a lower rate of severe knee injuries: 14.0% of knee injuries were serious, and 12.5% of all serious injuries were located at the knee joint. Thirty injuries were fractures, and of these, 26 (86.7%) were serious injuries. Sprains were the largest number of injuries (293), with 44 being serious. Of the 61 luxations, 35 (57.4%) were serious. Only 7.4% of all contusions were classified as serious. During the entire observation period, ten of the potentially very dangerous neck injuries had to be transferred to hospital. Four judokas had to be transferred to hospital after concussion/commotio cerebri. The short period of unconsciousness which occasionally occurs after strangling and choking techniques (“shime-waza”) was never a reason for transfer to hospital (Table 3).

Figure 1 shows that in male judokas, the number of injuries per weight category was distributed as to be expected with the number of participants in each weight category. In women (Figure 2), there was a remarkably high incidence of injuries in the under 52 kg category, and a low incidence in the under 57 kg category.

## 4. Discussion

The main findings of the present study is the higher frequency of injuries in female athletes, especially regarding injuries in the upper extremities in Europe’s top-level judokas during competitions over a period of 15 years. We realise that many injuries also occur during training, but this study was designed only to determine injury incidence during top-level tournaments. The mechanism of injury in judo is linked to throwing and grappling techniques. According to some studies, most injuries affect the upper limbs, as the fight starts with both judokas standing [10,14,17,18,19].

Lower limbs are at a high risk of injury as well [13]. In two studies on the Korean Olympic team judokas, the knee was frequently injured [13], with 20% of the injuries occurring in the trunk, especially in the lumbar and thoracic spine. These injuries occurred during training, not during competition. We found that 30% of the injuries occurred in the lower limbs (most at the knee), and 20.7% of injuries in the trunk and shoulder combined. Comparing judo and wrestling, the most common injuries were in the lower (judo 61%; wrestling 41%, *p* < 0.05) and upper limbs (judo 30%; wrestling 32%) [2]. In the present study, the knee (17.4%), shoulder (15.7%), and elbow (14.2%) were the primary anatomical locations of an injury.

Regarding injury types, in most studies, contusions and abrasions were the most frequent injuries. In the present investigation, sprains were the most frequent injury type, followed by contusion. Overall, 42.2% of all injuries were sprains, and 23.1% of all injuries were contusions, with no statistically significant difference between the genders. In other studies, sprains mainly occurred in the knee, elbow, and ankle, and often the judokas suffered sprains of the acromioclavicular (AC) joint [18,20]. Frey et al., evaluating judo competition-related injuries during 21 seasons in France, showed that the six most frequently sprained joints accounted for over 75% of total sprains [18]. Additionally, the incidence for overall sprain injuries was significantly higher in female athletes (0.82% vs. 0.53%, respectively; *p* < 0.001). Our study did not provide evidence of any differences between sexes (*p* > 0.05). The high rate of sprains, mainly the acromioclavicular joint, elbow, and knee, can be explained by falls on the shoulder or the use of the arm as a stabilizer in abduction to defend from a throwing attack [18]. Sprains of the knee or ankle are likely related to the rotational maneuvers required to attack and defend. In the present study, 44 of 293 sprains were considered serious, and the judokas had to be hospitalized. On the other hand, only 10 contusions, mostly a consequence of a fall, were serious.

The third major injury among judokas in the present study was a fracture, with 30 cases, 26 of which were serious, requiring transport to hospital. In four cases, the treating physician decided not to transport the injured athlete to hospital. These were finger fractures, and these injuries were likely treated after travelling back to the athlete’s home country. In the study of Frey et. al., clavicles were also the most commonly fractured bone [18], often from a direct fall onto the shoulder.

A major concern is cervical spine fracture, which can occur following hyperflexion or hyperextension of the cervical spine, or because of direct trauma or axial loading. Over the course of 15 years of injury recording, 10 neck injuries required transport to hospital.

In the present study, 47 of 695 medical interventions (6.8%) followed unconsciousness after a strangle/choke technique (shime-waza). In this case, judokas cannot stop the fight by themselves by tapping out, and the referee must immediately stop the fight. None of these were serious enough for the judoka to be transferred to hospital.

Concussions (commotio cerebri) were diagnosed only 19 times, and in four instances the judoka had to be transported to hospital [18,20,21,22].

There were no statistically significant differences in the occurrence of injury in the different weight categories, except a high injury rate in the women’s under 52 kg weight class. Rapid weight loss can impair the psychological and physiological performance of judokas [23], but our data collection system did not allow us to collate data in this respect. Lightweight judokas are more prone to elbow injuries, and heavyweight judokas are more prone to knee injuries.

A new approach toward motor abilities development in judo, including agility, coordination, foot work, strength, and explosive power of both the lower and upper limbs may reduce the occurrence of injury. Improved motor skills may allow to better control the exposure to full-body contact, decreasing the risk of injuries and increasing performance.

### 4.1. Limitations

We collected data on a large cohort of elite judokas over a relatively long period of time. However, we acknowledge that we do not have data on the outcome of these injuries, on their treatment beyond what was collected at the time of injury, and on the outcome. For example, we do not know whether some of these injuries required surgery, whether the injured athletes had to stop training and competing for any length of time beyond what happened at a given tournament, and when a judoka returned to training and competition. All these issues, despite the logistic efforts necessary to collect such data, should be the subject of future endeavours.

### 4.2. Application

The quantitative and qualitative monitoring of injuries sustained by contestants in high-ranking judo competitions is conducted by the EJU, aiming to use the data to develop and constantly review the rules of judo competitions. Thanks to the information obtained from this type of reports, the EJU has already modified the relevant regulations on several occasions. For example, some throws or defense against throws which exposed players to an increased risk of injury have now been ruled illegal.

## 5. Conclusions

The overall incidence of injuries during Europe’s high-level judo tournaments in the period 2005–2020 was 2.5%, with an incidence of serious injuries of 0.5%. Judo is therefore one of the Olympic sports with the lowest injury rate in competitions. The knee, shoulder, and elbow are the anatomical locations most prone to injury, with 20% of all injuries occurring in the upper limbs (including the shoulder) and 30% in the lower limbs. Sprains are the most frequent type of injury. There is no statistically significant difference between men and women in the overall injury rate, although women have significantly more elbow injuries. The women −52 kg weight category shows a remarkably higher injury incidence—this should be further investigated. It is remarkable that lightweights suffer more elbow injuries. Serious injuries are uncommon, and potentially very dangerous injuries, such as commotio cerebri and neck injuries, have a very low incidence rate. No judoka had to be transported to hospital after unconsciousness form strangulation/choking techniques.

### What Are the New Findings?

This study provides comprehensive data of injury rates and trends among male and female European elite judokas during international competitions.The women’s under 52 kg weight class has a remarkably high injury incidence.Female judokas experience significantly more elbow injuries than men.Judo has an overall injury incidence of only 2.5% during top-level tournaments in Europe.Potentially very serious injuries, like commotio cerebri or neck trauma, are uncommon.Unconsciousness after choking techniques has, in this study, never led to hospitalisation.

## Figures and Tables

**Figure 1 jcm-10-00852-f001:**
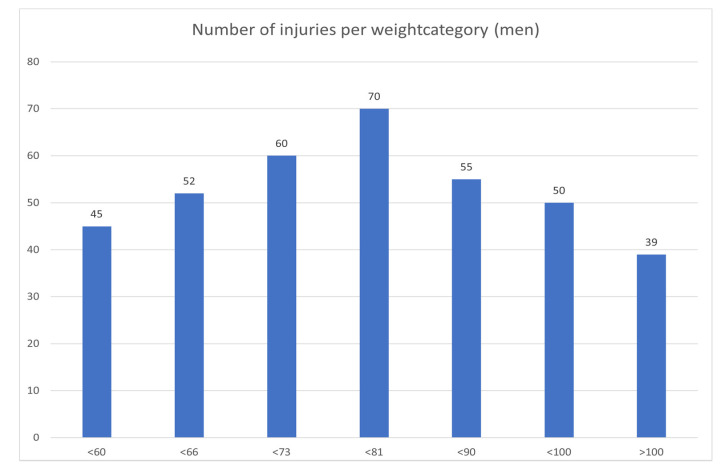
Distribution of injuries by weight category in men.

**Figure 2 jcm-10-00852-f002:**
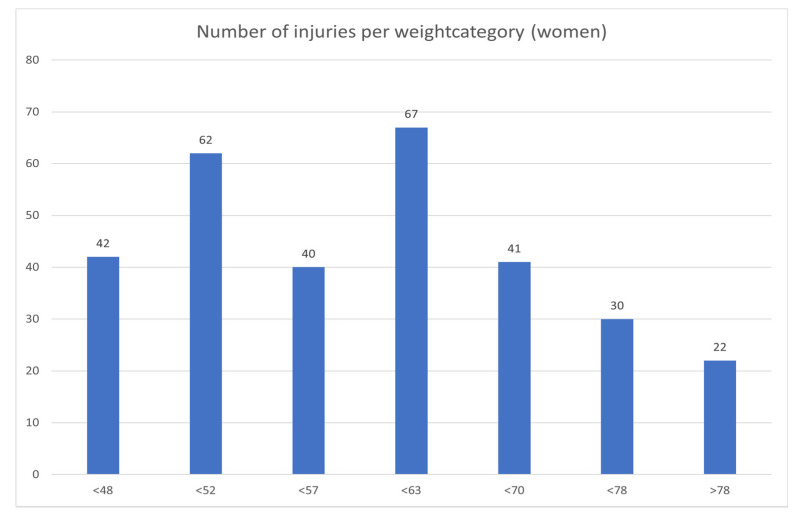
Distribution of injuries by weight category in women.

**Table 1 jcm-10-00852-t001:** Distribution of injuries by anatomical location.

Anatomical Location	Number	% of	Men	Women	Δ%	*p*
	of Injuries	Total Injury	No/%	No/%		
Head and neck						
Head	26	3.7	22 (3.2)	4 (0.6)	2.6	<0.05
Neck	34	4.9	21 (3.0)	13 (1.9)	1.1	>0.25
Eye	63	9.1	35 (5.0)	28 (4.0)	1	>0.25
Nose	9	1.3	6 (0.7)	3 (0.4)	0.3	N/a
Mouth	10	1.4	7 (1.0)	3 (0.4)	0.6	N/a
Throat	8	1.1	3 (0.4)	5 (0.7)	0.3	N/a
	14	2	5 (0.7)	9 (1.3)	0.6	<0.05
Upper body						
Trunk	24	3.4	14 (2.0)	10 (1.4)	0.6	>0.20
Shoulder	109	15.7	63 (9.1)	46(6.6)	2.5	>0.20
Back	11	1.6	6 (0.9)	5 (0.7)	0.2	N/a
Upper limb						
Elbow	99	14.2	44 (6.3)	55 (7.9)	1.6	<0.01
Hand	44	6.3	22 (3.2)	22 (3.2)	0	>0.25
Wrist	15	2.2	10(1.4)	5 (0.7)	0.7	>0.10
Lower limb						
Knee	121	17.4	60 (8.6)	61 (8.8)	0.2	>0.15
Ankle	38	5.5	19 (2.7)	19 (2.7)	0	>0.25
Foot	20	2.9	14 (2.0)	6 (0.7)	1.3	<0.05
Femur	17	2.4	11 (1.6)	6 (0.7)	0.9	>0.10
Calf	13	1.9	9 (1.3)	4 (0.6)	0.7	N/a
Others	21	3	11	10		
Total	696		382	314		

N/a—not applicable.

**Table 2 jcm-10-00852-t002:** Characteristics of the injuries.

Injury	Number of Injury	%	Men	Women	Δ%	*p*
			No/%	No/%		
Sprain	293	42.2	151 (21,7)	142 (20.4)	1.3	>0.15
Contusion	160	23.1	89 (12.8)	71 (10.2)	2.6	>0.25
Luxation	61	8.8	34 (4.9)	27 (3.9)	1	>0.25
Unconsciousness	47	6.8	21 (3.0)	26 (3.7)	0.7	>0.25
Bleeding	50	7.1	36 (5.2)	14 (2.0)	3.2	<0.01
Fracture	30	4.3	17 (2.4)	13 (1.9)	0.5	>0.25
Rupture	28	4	18 (2.6)	10 (1.4)	0.8	>0.25
Commotio cerebri	19	2.7	11 (1.6)	8 (1.2)	0.4	>0.25
Others	7	1	4 (0.6)	3 (0.4)	0.2	N/a
Total	695		381	314		

N/a—not applicable.

**Table 3 jcm-10-00852-t003:** Body areas and injuries classified as serious.

Anatomical Location	No of Injury	No of Serious Injury	% of Injuries in This Area Classified as Serious	% of All Serious Injuries
Shoulder	109	36	33	26.5
Elbow	99	32	32.3	23.5
Knee	121	17	14	12.5
Head	34	6	17.6	4.4
Ankle	38	6	15.8	4.4
Foot	20	6	30	4.4
Neck	63	10	15.9	7.4
Type of injury				
Sprain	293	44	15	32.3
Luxation	61	35	57.4	25.7
Fracture	30	26	86.7	19.1
Contusion	160	10	6.2	7.4
Rupture	28	9	32.1	6.6
Bleeding	50	5	10	3.7
Commotio cerebri	19	4	21	2.9
Unconsciousness	47	0	0	0

## Data Availability

The data presented in this study are available on request from the corresponding author.

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
