# Peer review of "Judo Injuries Frequency in Europe’s Top-Level Competitions in the Period 2005–2020"

_jcm, 2021, doi:10.3390/jcm10040852_

Round 1

Reviewer 1 Report

The aim of this study was to assess the prevalence of injury in Europe’s top-level judokas. Data were collected over a 15-year period through an injury registration form. With nearly 30000 competitors, the authors have collected a large amount of data, which is of great value for the judo associations worldwide.

Title

Given the use of a variety of concepts (prevalence, incidence, frequency), the authors are urged to decide on one term and use that throughout. Now, the title refers to “Identifying injury…” which is rather vague.

Abstract:

‘…define…’ should be ‘…defined…’

The sentence “There was no statistically…” seems odd given that there was a significant different re elbow injuries between men and women. Better to write with “There was a statistically significant…, but no other different between men and women.”

The authors mix the terms ‘prevalence’ and incidence’. Given that the aim was to assess prevalence, it seems out of place to include incidence in the conclusion; these two terms are epidemiology different.

Also, the Conclusion is merely a repetition of the Results, and therefore needs to be rewrittened.

Introduction

The authors report on data from the literature, some even longer than the present one. What is the gap in the literature? There is a great need for a focus on what is missing and the rationale for pursuing the present study, which then leads to the aim.

The aim is, however, not the same as in the Abstract where the authors write ‘prevalence’, whereas in the aim they refer to ‘frequency of injury’.

Materials and Methods

When was the ethical approval received? A year and possibly a number of the application should be given.

The definition of an injury could be given under subheading “Study design”, following the sentence that refer to ‘minor’ and ‘serious’ injuries.

I am somewhat surprised that the authors did not consider analysing trends in the injury reporting/frequency. With 15 years of data it should be possible to see if injuries are increasing or decreasing. I suggest that the authors add this to their manuscript.

Results

The injury type ‘contusion’, what exactly is referred to? Is that any type of a blow to any body part?

The term ‘unconsciousness’ is not a type of injury, rather a symptom of a brain injury. I assume that that it refers to Commotio cerebri (commonly known as concussion). I suggest that the authors use a different terminology and refer to head/brain injuries correctly, either with or without ‘unconsciousness’. The numbers may then seem more appropriate. Now it looks like  ‘unconsciousness’ is more common that ‘commotio’, which does not sense as they could both be related to a brain injury. When reading the text it appears that ‘unconsciousness’ refers to something else. I suggest the authors redefine ‘unconsciousness’ so that it is not mixed-up with ‘commotio’. Prhaps using fottnotes in the table could help?

There is some problem with the alignment, Rupture is followed by Commotio. Are they two separate injuries? If so, what is rupture?

In Table 3, Skull is usually used to define the cranium, ie bone. Do the authors mean Head?

Discussion

Please start this section with a summary of the main findings.

Line 163: this sentence does not make sense, please rewrite

Line 172: The authors state that contusions and abrasions were the most frequent injuries in the literature. The authors did not report these injuries, hence the difference. This should be noted here.

Line 194: Did any of the spine fracture lead to a spinal cord injury? If not, it would be important to report that as an SCI is a very serious injury.

Line 201: The authors report a rather low frequency of concussion. We know from other studies that this type of injury is underreported. It would be good to see here some elaboration of this. Is this an unexpectedly low number? Now, the sentence is merely a repetition of the results.

Limitation

As I pointed out earlier, it would have been interesting to see some kind of analyses of trends in injuries over this 15-year period.

The authors do not report on any strengths of the study. If their data collection method is deemed valid, a clear strength is that all injuires at this type of competition are reported which would give a rather nice base for discussing prevention.

I also miss some kind of practical implications of this study. What does the reporting lead to? I guess that not reporting on trends somewhat lower the importance of this study. If there is an increase over time, this calls for urgent preventive measures. This is also my main critique towards this study, which otherwise has several strengths.

Author Response

Title

Given the use of a variety of concepts (prevalence, incidence, frequency), the authors are urged to decide on one term and use that throughout. Now, the title refers to “Identifying injury…” which is rather vague.

 A: Thank you for your comment we have changed the title.

Abstract:

‘…define…’ should be ‘…defined…’

The sentence “There was no statistically…” seems odd given that there was a significant different re elbow injuries between men and women. Better to write with “There was a statistically significant…, but no other different between men and women.”

 A: Thank you for your comment we have modified the text.We changed the term prevalence with “frequency of injury”.

The authors mix the terms ‘prevalence’ and incidence’. Given that the aim was to assess prevalence, it seems out of place to include incidence in the conclusion; these two terms are epidemiology different.

Also, the Conclusion is merely a repetition of the Results, and therefore needs to be rewrittened.

 A: Thank you for your comment we have modified the text. We changed the term prevalence with “frequency of injury”.

Introduction

Regarding

The aim is, however, not the same as in the Abstract where the authors write ‘prevalence’, whereas in the aim they refer to ‘frequency of injury’.

 A: Thank you for your comment we have modified the text. We changed the term prevalence with “frequency of injury”. 

Materials and Methods

When was the ethical approval received? A year and possibly a number of the application should be given.

 A: There is addition of theprotocole number of the ethical approval.

Results

The injury type ‘contusion’, what exactly is referred to? Is that any type of a blow to any body part?

The term ‘unconsciousness’ is not a type of injury, rather a symptom of a brain injury. I assume that that it refers to Commotio cerebri (commonly known as concussion). I suggest that the authors use a different terminology and refer to head/brain injuries correctly, either with or without ‘unconsciousness’. The numbers may then seem more appropriate. Now it looks like  ‘unconsciousness’ is more common that ‘commotio’, which does not sense as they could both be related to a brain injury. When reading the text it appears that ‘unconsciousness’ refers to something else. I suggest the authors redefine ‘unconsciousness’ so that it is not mixed-up with ‘commotio’. Perhaps using fottnotes in the table could help?

 A: Thank you for your comment. There is addition of definition of contusion and unconsciouness in the manuscript.

In Table 3, Skull is usually used to define the cranium, ie bone. Do the authors mean Head?

 A: Yes, the word skull it is replaced with head injuries.

Discussion

Please start this section with a summary of the main findings.

 A: I would suggest addition of a paragraph, from the first author, with the main findings (higher frequency of injuries in female athletes and regarding injuries in upper extremities)

Line 163: this sentence does not make sense, please rewrite

 A: Many Thanks to the reviewr .We did

Line 172: The authors state that contusions and abrasions were the most frequent injuries in the literature. The authors did not report these injuries, hence the difference. This should be noted here.

Line 194: Did any of the spine fracture lead to a spinal cord injury? If not, it would be important to report that as an SCI is a very serious injury.

Line 201: The authors report a rather low frequency of concussion. We know from other studies that this type of injury is underreported. It would be good to see here some elaboration of this. Is this an unexpectedly low number? Now, the sentence is merely a repetition of the results.

 A: Thank you for your comment we consider this in ‘Limitations’ section.

Limitation

As I pointed out earlier, it would have been interesting to see some kind of analyses of trends in injuries over this 15-year period.

The authors do not report on any strengths of the study. If their data collection method is deemed valid, a clear strength is that all injuires at this type of competition are reported which would give a rather nice base for discussing prevention.

 A: there was reporting of all injuries in all the competitions with the same standardized procedure.

I also miss some kind of practical implications of this study. What does the reporting lead to? I guess that not reporting on trends somewhat lower the importance of this study. If there is an increase over time, this calls for urgent preventive measures. This is also my main critique towards this study, which otherwise has several strengths.

A: The quantitative and qualitative monitoring of injuries sustained by contestants in high-ranking judo competitions is conducted by the EJU with the aim of using this data in the development and revision of judo sports rules. Thanks to the information obtained from this typeof reports, the EJU has already modified the sports regulationsseveral times, which, for example, deleted some throws or defenseagainst throws, which exposed players to an increased risk of injury. For example, after Olympic Games in Rio (2016), it was prohibited to use back fall protection by making a “bridge” over the head by a throwing competitor (all situations of voluntarily landing in the “bridge position”, will be considered hansoku-make- disqualification). A competitor who does not comply with this rule will be disqualified from participating in thetournament.

Reviewer 2 Report

I put all my comments in the pdf file.
The paper would have more scientific value if you had more accurate diagnoses.

Author Response

Dear reviewer,

Thank you for reviewing our manuscript. We made a modification in the text and the replies were added to your comments in PDF

Round 2

Reviewer 1 Report

The authors have responded to all my comments and queries. However, the update of the text is in many places not grammatically correct, and I urge the authors to have the paper proof read appropriately by a native speaking person.